# Wavelength-Tunable Narrow-Linewidth Laser Diode Based on Self-Injection Locking with a High-Q Lithium Niobate Microring Resonator

**DOI:** 10.3390/nano13050948

**Published:** 2023-03-06

**Authors:** Ting Huang, Yu Ma, Zhiwei Fang, Junxia Zhou, Yuan Zhou, Zhe Wang, Jian Liu, Zhenhua Wang, Haisu Zhang, Min Wang, Jian Xu, Ya Cheng

**Affiliations:** 1The Extreme Optoelectromechanics Laboratory (XXL), School of Physics and Electronic Science, East China Normal University, Shanghai 200241, China; 2State Key Laboratory of High Field Laser Physics and CAS Center for Excellence in Ultra-Intense Laser Science, Shanghai Institute of Optics and Fine Mechanics (SIOM), Chinese Academy of Sciences (CAS), Shanghai 201800, China; 3Center of Materials Science and Optoelectronics Engineering, University of Chinese Academy of Sciences, Beijing 100049, China; 4Hefei National Laboratory, Hefei 230088, China; 5State Key Laboratory of Precision Spectroscopy, East China Normal University, Shanghai 200062, China; 6Collaborative Innovation Center of Extreme Optics, Shanxi University, Taiyuan 030006, China; 7Collaborative Innovation Center of Light Manipulations and Applications, Shandong Normal University, Jinan 250358, China

**Keywords:** lithium niobate, microring resonator, photolithography, chemo-mechanical etching, narrow linewidth, self-injection locking

## Abstract

We demonstrate a narrow linewidth 980 nm laser by self-injection locking of an electrically pumped distributed-feedback (DFB) laser diode to a high quality (Q) factor (>10^5^) lithium niobate (LN) microring resonator. The lithium niobate microring resonator is fabricated by photolithography-assisted chemo-mechanical etching (PLACE) technique, and the Q factor of lithium niobate microring is measured as high as 6.91 × 10^5^. The linewidth of the multimode 980 nm laser diode, which is ~2 nm measured from its output end, is narrowed down to 35 pm with a single-mode characteristic after coupling with the high-Q LN microring resonator. The output power of the narrow-linewidth microlaser is about 4.27 mW, and the wavelength tuning range reaches 2.57 nm. This work explores a hybrid integrated narrow linewidth 980 nm laser that has potential applications in high-efficient pump laser, optical tweezers, quantum information, as well as chip-based precision spectroscopy and metrology.

## 1. Introduction

The narrow-linewidth laser sources are widely used in frontier science and technology, such as coherent optical communication [1], optical atomic clock [2,3], spectral measurement [4], gravitational wave measurement [5], and other fields. Self-injection locking is an effective scheme to generate narrow-linewidth laser from electrically pumped laser diode by resonant optical feedback of external optical cavity, which is simple, compact, and cheap [6]. Self-injection locking can provide a single longitudinal mode, narrow linewidth, stable frequency output, as well as high tuning efficiency and large tuning range [7,8,9,10,11,12,13,14,15,16,17,18,19,20,21,22,23,24,25,26]. Self-injection locking of an on-chip laser diode to high quality (Q) factor optical microresonator results in sub-Hz to kHz linewidth, which is several orders of magnitude smaller than the original linewidth of the free-running laser diode [7,8,9]. Optical resonators can be high finesse Fabry–Pérot (FP) cavities and whispering gallery mode (WGM) microresonators [10,11,12,13,14]. The high-finesse FP resonators have been widely used but are comparatively bulky [10,11]. In contrast to the FP cavities, the WGM microresonators can be easily incorporated with laser diodes due to the chip-scale sizes and ease of integration [12,13,14]. Recently, the hybrid integrated narrow linewidth laser realized by coupling a laser diode chip with a high-Q microresonator chip has been studied extensively and made rapid progress [6]. The high-Q WGM microresonators are fabricated on many material platforms, including Si, MgF_2_, CaF_2_, and Si_3_N_4_ [15,16,17,18,19,20,21]. Lithium niobate (LN) is an attractive material platform for integrated photonics applications and has been widely used for many decades due to its wide transparency window, high refractive index, as well as large acusto-optic, electro-optic, thermal-optic, and nonlinear optical coefficients [27,28]. Combining low propagation loss with large electro-optical coefficient, the LN WGM microresonators have been used as the self-injection locking component to achieve kilohertz linewidth laser output and record high-frequency modulation speed up to exahertz/s [22,23,24]. However, most of the self-injection locking research focuses on the optical communication wavelength band around 1550 nm. Recently, self-injection locked narrow-width lasers have extended their operation wavelengths to the mid-infrared and visible bands [25,26]. However, to date, the 980 nm narrow linewidth laser by self-injection locking has not been demonstrated.

The reliable semiconductor 980 nm pump lasers are key components in ytterbium/erbium ions doped lasers and optical fiber amplifiers, which are critical to high-throughput laser material processing and optical fiber communication systems [29,30]. Pumping by 980 nm is advantageous due to chip-scale size, high-power, inherent low amplifier noise, and the high electrical-to-optical conversion efficiency. The narrow linewidth, stable, and high-power lasers need to operate at low-temperature conditions, so the absorption spectrum of laser working media becomes very narrow. Therefore, it needs the narrow-linewidth and precisely tunable pump lasers around 980 nm. In addition to pump laser sources, the 980 nm can also apply to optical tweezers and new blue-green laser sources [31,32]. Near-infrared laser beams are generally used to trap small objects and cells with small thermal effect and freely without photo-damage to biological samples. The narrower the spectrum and the deeper the potential well generated by the optical force, the smaller the particles that can be trapped. Therefore, the high-power narrow-linewidth wavelength-tunable 980 lasers have considerable potential for applications in optical tweezers. The narrow-wavelength blue-green laser sources are an important laser source for high-resolution spectroscopy, optical clocks, atomic cooling, and trapping, as well as quantum information [33,34,35]. The high-power narrow-linewidth tunable 980 nm lasers are attractive to build new blue sources around 490 nm by the second frequency generation (SHG) process. Furthermore, the high-power narrow-linewidth 980 nm lasers have applications in LIDAR, precision spectroscopy, and metrology.

Here, a high-performance narrow-linewidth 980 nm laser has been demonstrated by self-injection locking of a seed laser to a high-Q lithium niobate (LN) microring resonator. The 980 nm laser diode source is a commercial Chip-on-Submount (CoS) laser. The linewidth of 980 nm COS laser diode is reduced from about 2 nm to 35 pm. We also observed the transition from multimode to single-mode operation by monitoring the lasing modes at the output of the LN microring resonator using an optical spectral analyzer. The output power of the narrow-linewidth laser is about 4.27 mW, and the narrow-linewidth laser can be tuned in a wavelength range of 2.57 nm. In this self-injection locking scheme, the LN microring is fabricated by the photolithography-assisted chemo-mechanical etching (PLACE) technique developed in our lab, whereas other high-Q WGM microresonators used in self-injection locking schemes are often fabricated by electron beam lithography (EBL) [19,20,21]. It is well-known that the EBL has a limited writing field of ~1 mm^2^ and a low writing speed of ~1 cm^2^/hour, whereas the PLACE technique has a large writing field (8 inch) and a high writing speed (1 cm^2^/min), so the PLACE technique can easily fabricate the 4-inch LNOI wafer in 2 h in our lab. The footprint of a photonic chip fabricated by PLACE technique is only limited by either the wafer size or the motion range of the positioning stage, due to the unique advantage of femtosecond laser direct writing. Thus, the PLACE technique allows for the expansion of the footprint of photonic chip almost up to 0.5 × 0.5 m. In addition, the photonic structures fabricated by the chemo-mechanical polishing (CMP) process can have a surface roughness sub-nanometer, reducing the propagation loss approaching the absorption limit of the LN crystal [36]. These unique advantages are critical for developing large-scale photonic chips [37].

## 2. Experimental Details

Figure 1a–f shows the illustration schematic of the fabrication process of an on-chip high-Q LN microring resonator by PLACE technique, which mainly consists of the following procedures. Firstly, a commercially available thin film lithium niobate on insulator (TFLNOI) wafer from Jinan Jingzheng Electronics Co., Ltd., (Jinan, China) was cut into square (4 mm × 7 mm) samples by a wafer dicing machine. As shown in Figure 1a, the TFLNOI sample had a stack of X-cut thin film lithium niobate (TFLN)/buried oxide (SiO_2_)/Silicon (Si) where the thicknesses of X-cut TFLN, SiO_2_ and Si were 500 nm, 4.7 µm and 525 µm, respectively. Secondly, as shown in Figure 1b, a 200 nm thick chromium (Cr) film was coated on the surface of the TFLNOI sample by a magnetron sputtering system (ULVAC Inc., Chigasaki, Japan). Thirdly, as shown in Figure 1c, the Cr film was patterned into a microring coupled with a bus waveguide as hard masks using space-selective femtosecond (fs) laser ablation (Pharos-6W: 1030 nm, Light conversion, Vilnius, Lithuania). In this step, the femtosecond laser beam was focused on a spot with a diameter of 1.8 μm using an objective lens (100×/NA 0.7 Mitutoyo, Kanagawa, Japan) the ultrafast laser processing has a strong threshold effect of improving the resolution of 500 nm beyond the diffraction limit [38]. Meanwhile, the translation of the TFLNOI sample was controlled in 3D space with 100 nm resolution using an air-bearing XYZ nanopositioning stage (ABL15020 Aerotech, Pittsburgh, PA, USA). Fourthly, as shown in Figure 1d, the laser-written Cr hard mask was transferred to the TFLN below by the chemical-mechanical etching (CMP) process by a wafer polishing machine (NUIPOL 802, Kejing, Inc. Shenyang, China). We chose Cr as mask because the hardness of Cr was much higher than lithium niobate. In the CMP process, a piece of a polishing pad and the colloidal silica polishing suspension (MasterMet, Buehler, Ltd. , Lake Bluff, IL, USA) were used. It should be noted that in the microring resonator structure, it is relatively difficult to inject the suspended colloidal silica spheres into the narrow groove coupling area between the microring and the bus waveguide. If the coupling distance between the waveguide and the microring is too small, the suspended colloidal silica spheres cannot be injected into the groove and contact the surface of TFLN for etching. Therefore, the coupling distance between the waveguide and the microring should be controlled wide enough to be polished. Fifthly, the sample after CMP process is shown in Figure 1e, the ports on both sides of the sample were then polished with a polishing solution containing cerium oxide polishing powder for smooth edges. Under the premise of protecting the lithium niobate structure under the mask to the greatest extent, the edge of the lithium niobate microring was polished smooth. Finally, as shown in Figure 1f, the Cr hard mask is chemically etched with chromium etching solution (Alfa Aesar, Haverill, MA, USA), and the sharp part at the top of the microring resonator and waveguide were removed with a secondary-slight CMP without hard mask. The LN microring chip was prepared and ready to be integrated with the 980 nm laser diode.

## 3. Results and Discussion

Figure 2a shows a micrograph of the fabricated LN microring resonator with a circumference of about 2 mm and a curvature radius of 500 µm designed to reduce bending losses. The LN microring is also designed with improved quality factors using quarter Bezier curves due to mode match in coupling region and lower bending loss than the conventional circular ring resonator. Figure 2b is a close-up of one half of the coupling region. The different thicknesses of the bottom of the gap and the outer edge of the waveguide can be clearly seen through the color changes in the optical micrograph of the coupling region. As it is relatively difficult to inject the suspended colloidal silica spheres into the narrow groove coupling region mentioned above, the coupling distance between the straight waveguide and the microring should be controlled at 5.8 μm, which is wide enough to be polished. Meanwhile, in order to better couple the input light into the microring, the coupling length should be increased to 115 µm to improve the coupling efficiency of the straight waveguide and the ring for 980 nm wavelength, which is much shorter than the C-band. Figure 2c shows a close-up of the end facet of the waveguide after end polishing, smooth end face can reduce the coupling gap and improve coupling efficiency. To reduce light scattering loss, the waveguide is designed to be a ridge structure, and the TFLN is only partially etched to preserve the ridge on the TFLN slab. Figure 2d is the false-color scanning electron microscope (SEM) image of the cross-section of the waveguide and microring fabricated by the PLACE technique. Figure 2e shows the dimension annotation of the cross-section of the waveguide, where $w_{0}=1\;\mu m$,${\;w}_{1}=1.14\;\mu m$,${\;w}_{2}=4.8\;\mu m$,${\;T}_{0}=90\;\mathrm{nm}$,${\;T}_{1}=120\;\mathrm{nm}$,${\;T}_{2}=290\;\mathrm{nm}$. The surface roughness of the waveguide was measured by atomic force microscopy (AFM) to be sub-nanometer in our previous work [39].

As shown in Figure 3a, to characterize the transmission spectrum and Q factor of the fabricated LN microring resonator, a continuously tunable laser (CTL 980, TOPTICA Photonics Inc. Munich, Germany) is input into the microring chip through a lensed fiber (CXFIBER Inc.). The focused spot of lensed fiber is reduced to about 2 µm. The output signal is collected by the same lens fiber and input into the photodetector (New focus 1811 Newport, Irvine, USA) and then connected to the oscilloscope (OSC, Tektronix MOS 64, Beaverton, OR, USA) to monitor the coupling condition of the resonant mode for the measurement of Q-factor of the LN microring. Meanwhile, the polarization state of the laser is controlled by the in-line fiber polarization controller (FPC) to achieve the optimal coupling of the resonant mode in the microring. Moreover, an arbitrary function generator (AFG3021C, Tektronix, Beaverton, OR, USA) is used to generate an external drive triangular wave electrical signal (100 Hz, 2 V_pp_) applied to the continuously tunable laser (CTL) to scan finely around 980 nm wavelength. At the same time, the electrical signal from the AFG was also linked to the OSC to trigger and synchronize the transmission spectrum channel. Figure 3b,c shows the two adjacent resonant modes in the normalized transmission spectra of the microring and their Lorenz fitting curves (red). The two sets of loaded Q factors measured by the microring laser around the resonance wavelength of 975 nm (pump laser) are 6.91 × 10^5^ (975.36 nm) and 6.53 × 10^5^ (975.51 nm), respectively. The free spectral range (FSR) of the LN microring is 0.15 nm.

Figure 4a shows the illustration schematic of the 980 nm narrow linewidth self-injected locking laser consisting of a commercial CoS laser diode and the high-Q LN microring. The emitting spot size of CoS 980 nm laser diode is 4 × 1 μm, the beam divergence angle parallel is 8°, and perpendicular is 40°, respectively. The input size of LN microring taper waveguide is 4 × 0.5 μm. The coupling efficiency is about 10 dB/facet due to the mismatch between the spot size and the Fresnel reflection at the end face. Figure 4b shows the top view of the fabricated self-injected narrow linewidth laser, two probes (ST-20-0.5, GGB Industries Inc. Naples, FL, USA) to apply direct current (DC) on the electrodes of CoS laser. As shown in Figure 4c, the input port of the LN microring should be aligned precisely with the output port of the CoS laser diode to achieve efficient optical coupling. This is ensured using a 6-axis alignment system with a resolution of 10 nm and monitored by a microscopic imaging system in real time.

As shown in Figure 5a, to characterize the performance of the self-injection locking laser, we select a power meter (New focus 1811-FC-AC, Newport Inc. Irvine, CA, USA) and an optical spectral analyzer (OSA: AQ6375B, YOKOGAWA Inc. Tokyo, Japan) to analyze the output spectrum. Figure 5b is the comparison of the laser linewidth for the free-running DFB laser diode and the self-injection locking laser. As shown in Figure 5b, the two peak laser emission in blue is the multimode laser form free-running 980 nm DFB laser. Multiple peak fitting is carried out for the two peak laser spectra, and the two modes are, respectively, the green curve and the orange curve in Figure 5b. The line width is 2000–3000 pm, which is consistent with the factory specification. The Qs of the laser diode cavity at 978.79 and 982.47 nm measured are $3.6\times 10^{2}$ and $2.75\times 10^{2}$, respectively. As shown in narrow-linewidth laser emission at 982 nm, the transition from multimode to single mode can be observed clearly after self-injection locked with LN microring resonator. The DFB laser diode is forced to oscillate at the cavity resonant frequency in the self-injection locking condition. Assuming that the frequency noise of the laser is white, the frequency noise suppression ratio is
(1)$$\frac{\delta \omega }{\delta \omega_{free}}\approx \frac{Q_{d}^{2}}{Q^{2}}\times \frac{1}{16\Gamma_{m}^{2}\left(1+\alpha_{g}^{2}\right)},$$
where *δω* is the linewidth of self-injection-locked DFB laser, *δω_free_* is the linewidth of the free-running DFB laser, *Q_d_* (10^2^) and *Q* (10^5^) are the quality factors of the laser diode cavity and of the LN microring cavity, and *α_g_* is the phase-amplitude coupling factor. According to the formula, the order of magnitude of the linewidth compression factor is 10^6^, and thus the theoretically calculated linewidth is 600 kHz (1.92 × 10^3^ pm). However, the experimentally measured linewidth is 35 pm, which is much wider than the theoretical prediction. Actually, the measurement resolution of the linewidth of our self-injection locked laser diode is limited by the highest resolution of OSA, which is only 0.01 nm. The actual linewidth compression factor should be much greater than that measured by OSA, and the actual linewidth needs to be characterized with the self-delayed heterodyne system, which is currently not available in our laboratory. Such a measurement will be carried out in the future.

As shown in Figure 6a, we measured a maximum output power of 4.27 mW for the narrow linewidth self-injected locking laser operation at 982 nm wavelength. Figure 6b shows the tuning of the narrow linewidth laser wavelength by changing the applied electrical power of the DFB laser. The tuning process is easy to realize and control at low applied electrical power, while the tuning process becomes unstable at high power, i.e., the operation wavelength fluctuates at high output power. The output lasing wavelength redshifts with the increase of the applied electrical power in the tuning range of 2.57 nm. It should be noted that the lasing wavelength is tuned stepwise but not continuously in our experiment. Further improvement by the integration of microelectrodes near the LN microring resonator will enable continuous tuning. Combining the stepwise rough-tuning with continuous fine-tuning allows our 980 nm narrow linewidth self-injected locking laser to have a wide tuning range spanning over many FSRs with high tuning resolution with an FSR.

## 4. Conclusions

In conclusion, we demonstrate a high-performance narrow linewidth 980 nm laser by self-injection locking of a laser diode to a LN microring resonator of high-Q. The LN microring resonator is fabricated by PLACE technique which has a high-scale writing field, high writing speed, and sub-nanometer surfaces. The Q factor of lithium niobate microring is measured as high as 6.91 × 10^5^. An OSA is used to characterize the linewidth of self-injection locked laser diode, which is narrowed down from about 2 to 35 pm (limited by the resolution of OSA). The transition from multimode to single mode can be observed clearly after a self-injection locked with LN microring resonator. The output power from the integrated narrow-linewidth laser diode is 4.27 mW, and the wavelength is tuned with a range of 2.57 nm. This narrow-linewidth 980 nm laser has potential applications in high-efficient pump lasers, optical tweezers, quantum information, as well as chip-based precision spectroscopy and metrology. Further improvement by fabrication of the spot size converter (SSC) on the LN microring chip both for lensed fiber and DFB laser, the spot size of input waveguide on LN microring chip must match the DFB laser diode, and the spot size of output waveguide on LN microring chip must match the fiber and as closely as possible to ensure good coupling. We can integrate microelectrodes near the LN microring resonator will enable continuous precious and high-speed tuning by electro-optic and thermal-optic effects in LN crystal. We will mount the self-injection locked laser on a thermoelectric refrigerating unit (TEC) for more stable laser emission, and TO-packaged can also be applied to the self-injection locked laser chip. The SiO_2_ cladding will be coated on LN microring chip to prevent contamination for stable high Q. More importantly, we will measure the linewidth accurately using the delayed self-heterodyne interferometer setup or high-quality laser linewidth/phase noise analyzer in the future.

## Figures and Tables

**Figure 1 nanomaterials-13-00948-f001:**
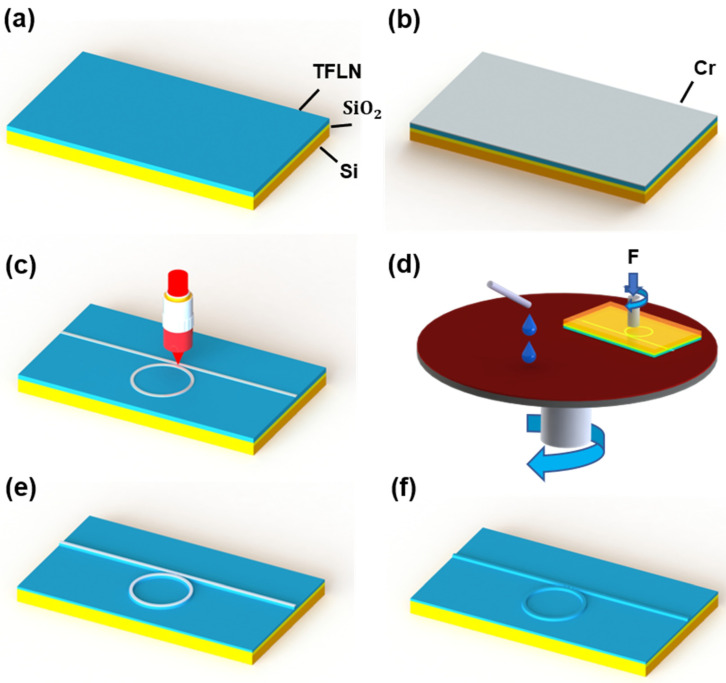
(**a**–**f**) Illustration of the on-chip LN microring resonator fabrication. (**a**) TFLNOI wafer consisted of LN, SiO_2_ layer, and silicon substrate. (**b**) A 200 nm-thick Cr film was coated on the surface of the TFLNOI wafer. (**c**) The Cr pattern was generated by femtosecond laser direct ablation. (**d**) The laser-written Cr mask was transferred to the TFLN below by the CMP process. (**e**) The edge of the lithium niobate microring was polished smooth after CMP polishing. (**f**) The Cr hard mask was chemically etched with a chromium etching solution.

**Figure 2 nanomaterials-13-00948-f002:**
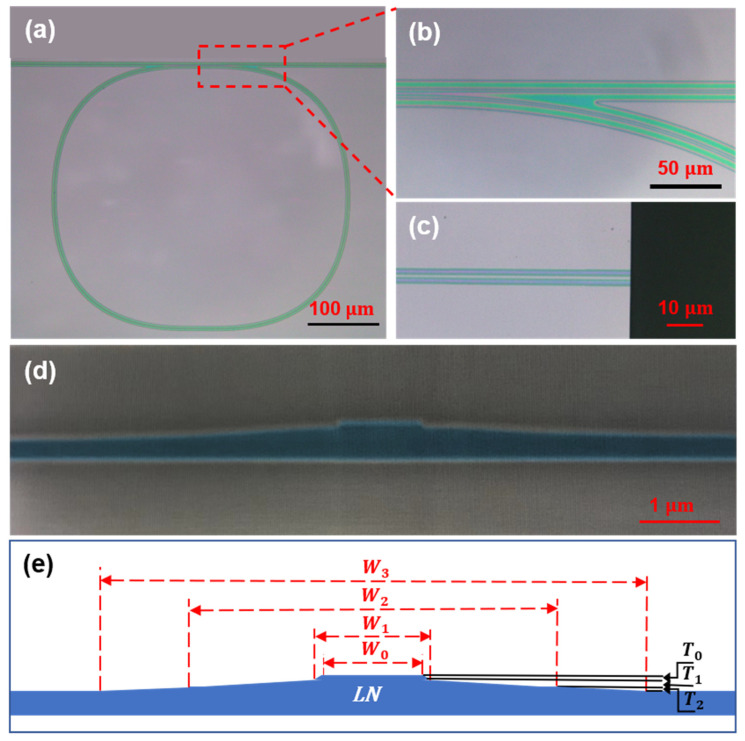
(**a**) Microscope image of the proposed LN microring. (**b**) Close-up of half of the coupling region. (**c**) Close-up of the end facet of the waveguide. (**d**) The false color scanning electron microscope (SEM) images of the cross-section of the waveguide and microring. (**e**) Illustration schematic of ridge waveguide structure on TFLN fabricated by PLACE technique.

**Figure 3 nanomaterials-13-00948-f003:**
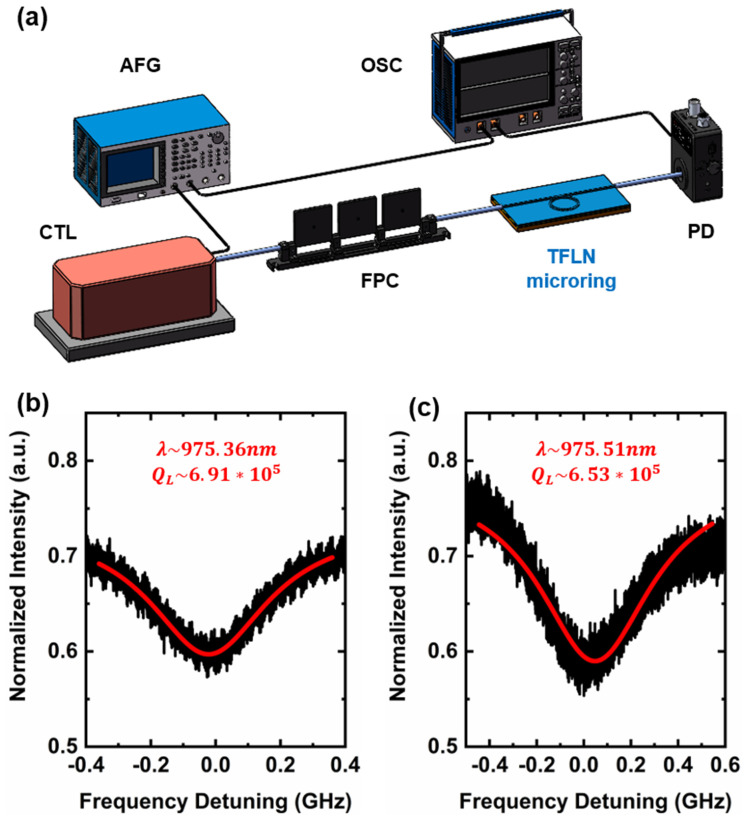
(**a**) Schematic diagram of the experimental apparatus for the characterization of the loaded Q factor of the LN microring resonator (CTL: continuously tunable laser; AFG: arbitrary function generator; FPC: fiber polarization controller; PD: photodetector; OSC: oscilloscope). (**b**,**c**) The two adjacent transmission spectra of the LN microring laser at 975 nm of the pumped laser. The Lorentz fitting (red curve) reveals a Q-factor of 6.91 × 10^5^ at the wavelength of 975.36 nm, while the Lorentz fitting (red curve) reveals a Q-factor of 6.53 × 10^5^ at the wavelength of 975.51 nm.

**Figure 4 nanomaterials-13-00948-f004:**
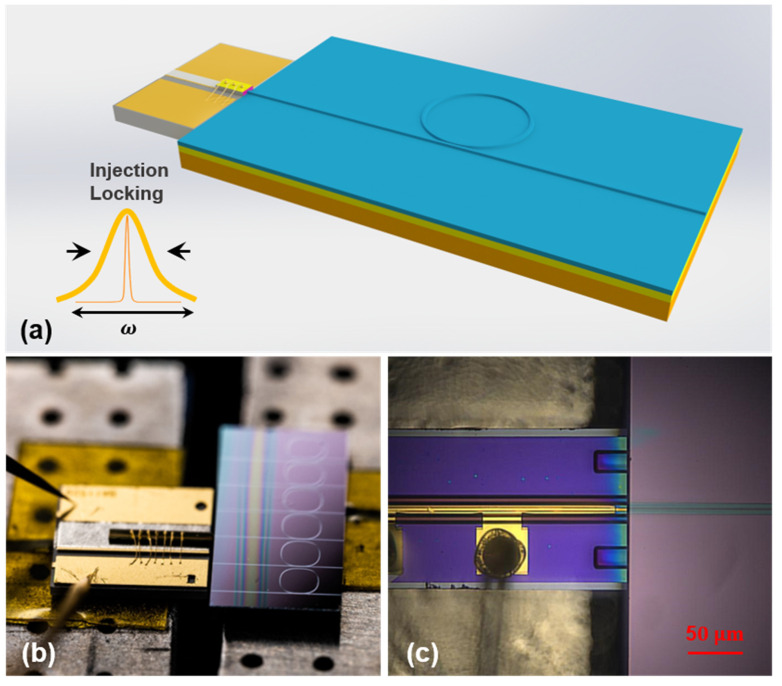
(**a**) Illustration scheme of the narrow linewidth self-injection-locked laser, which is composed of a commercial CoS laser diode and a high-Q LN microring laser. (**b**) A photograph of the proposed self-injection-locked laser is taken by a camera. (**c**) Close-up optical micrograph of the interface between the CoS laser diode and LN microring.

**Figure 5 nanomaterials-13-00948-f005:**
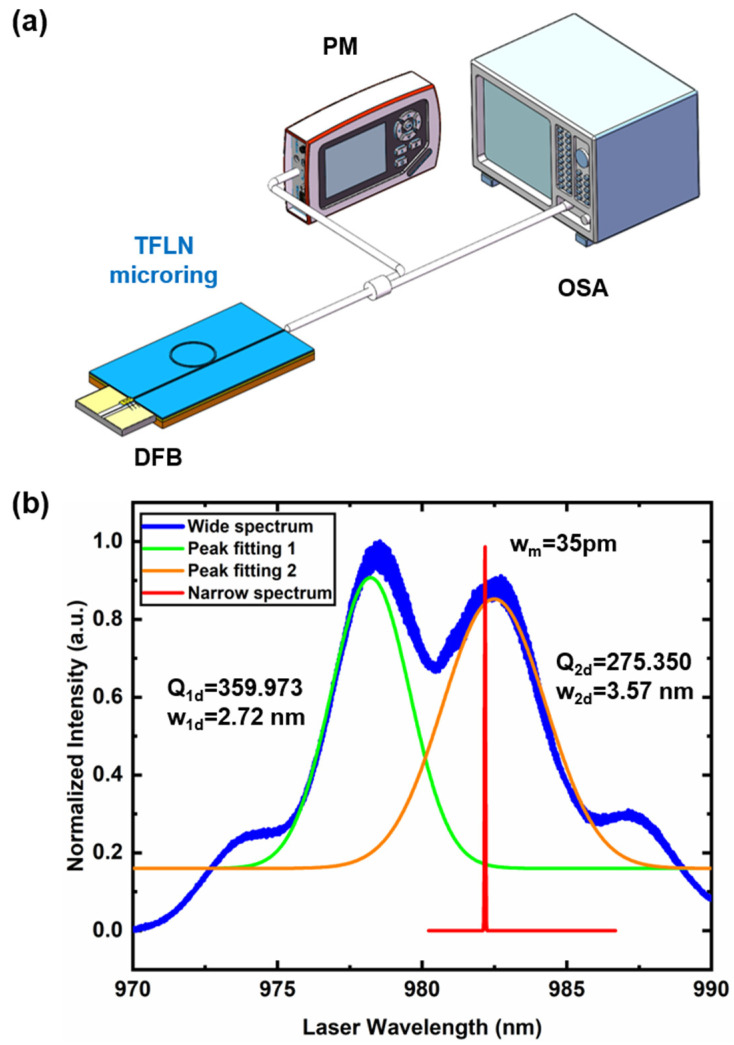
(**a**) Schematic diagram of the experimental setup of the self-injection locking narrow linewidth 980 nm laser (DFB: Distributed-feedback laser; PM: Power meter; OSA: Optical spectrum analyzer). (**b**) Comparison of the laser linewidth for the free-running DFB case and the case where the DFB is self-injection-locked to an LN microring cavity.

**Figure 6 nanomaterials-13-00948-f006:**
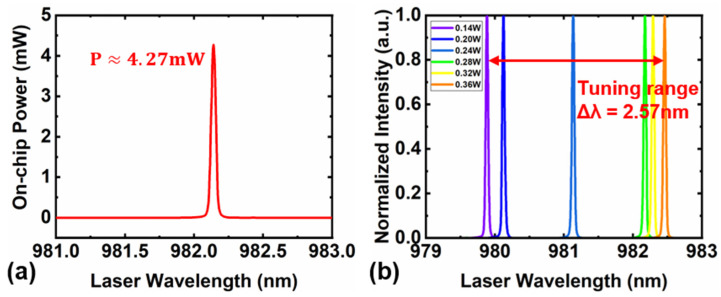
(**a**) Spectral power density of 980 nm narrow linewidth self-injected locking laser emission. (**b**) The dependence of the narrow linewidth laser wavelength on the applied electrical power.

## Data Availability

Data underlying the results presented in this paper are not publicly available at this time but may be obtained from the authors upon reasonable request.

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
