# Peer review of "Wavelength-Tunable Narrow-Linewidth Laser Diode Based on Self-Injection Locking with a High-Q Lithium Niobate Microring Resonator"

_nanomaterials, 2023, doi:10.3390/nano13050948_

Round 1

Reviewer 1 Report

The manuscript presents a description of development of 980-nm narrow-linewidth laser using resonant optical feedback of external optical cavity. The manuscript is written in a logical and understandable language. The reliability of the obtained results is beyond doubt. It may be of interest to specialists in photonics, optical communications, spectroscopy, etc.

The authors believe that their main merit is that they managed to create a narrow-band laser with a wavelength of 980 nm, which was not previously realized for this type of lasers. In this regard, I would like to make the following remark. In itself, the creation of a laser with a wavelength slightly different from analogues is not a sufficient basis for publication in the journal Nanomaterials. Therefore, the authors should focus on describing the fundamental problems that needed to be solved in order to master their chosen spectral range or on fundamental differences in the scheme or technology for creating the laser they describe.

Author Response

The author response can find in the attached file.

Reviewer 2 Report

In this work, the authors experimentally demonstrated a tunable laser diode response controlled by a ring resonator. The authors achieve a very narrow linewidth, and the response is tuned by controlling the pump power. The works present an alternative for coupling laser diodes; here, the traditional stabilized are FBGs. As a result, the manuscript is original, and the results are suitable for practical application. However, the authors need to validate and clarify some points before acceptance. 

*The thermal analysis is not included; what is the temperature operation? Moreover, what does the wavelength shifting generated by the temperature?

*The authors need to include the stability analysis. What are the wavelength and power fluctuations?

* The authors need to mention the effect of the roughness. This feature affects the performance of the device and the coupling efficiency. Please include a discussion. 

*A lateral microscope face view of the platform ring resonator is required; Here, the authors need to discuss the nanofabrication challenges. Moreover, how can the performance be affected by incorporating cladding? 

*A numerical aperture discussion needs to include understanding the coupling laser diode. 

*What is the high Q factor value? Please include the Q factor analysis to validate the statement “..microring resonator of high-Q.”

*Figure 4 needs a line color description. Please include a proper caption. 

Author Response

(The authors gave the same response as above.)

Reviewer 3 Report

Comments on:

 Wavelength-tunable narrow-linewidth laser diode based on self-injection locking with a high-Q lithium niobate microring resonator

by Ting Huang et al

The paper presents a possibility to build a narrow linewidth laser with high power and high performance. The paper is well written and easy to comprehend. I have only a few minor textual comments that should be straightforward to implement in the new version. I definitely recommend this study for the publication.

L 31: 4.2 mw -> 4.2 mW . Same for many other places, l73, 181 and others

L64-66: “Yet to date, the 980 nm … and laser pump source” The message of this sentence is not clear to me. Please, re-phrase or rewrite it

Figure 1 caption:  “(a)(d) Flowchart of fabricating a on-chip LN microring resonator ” -> Illustration of the on-chip LN microring resonator fabrication.

L141-151: Please, use the same units for the same physical quantities. You use kHz, pm and nm for the linewidth. It is quite difficult to follow and compare.

Figure 4: Please, explain the colours and different curves in the caption.

Author Response

(The authors gave the same response as above.)

Reviewer 4 Report

The review of the article you  can find in the attached file.

Author Response

Thanks to the reviewer for this nice question. Firstly, we added the description of the fundamental problems be solved in our chosen spectral range and fundamental differences in the scheme or technology for creating our self-injection locking laser in introduction part. Secondly, we cited more relevant literatures and rewrote the conclusions. Finally, we also reorganized the paper including figures and text as shown in the revised manuscript. 

Round 2

Reviewer 1 Report

The new edition of the manuscript looks much better than the previous one. It may be published in its present form.